# Sparse Autoencoders Find Highly Interpretable Features in Language Models

**Hoagy Cunningham**[*12], **Aidan Ewart**[*13], **Logan Riggs**[*1], **Robert Huben**, **Lee Sharkey**[4]
[1]EleutherAI, [2]MATS, [3]University of Bristol, [4]Apollo Research
{hoagycunningham, aidanprattewart, logansmith5}@gmail.com

## ABSTRACT

One of the roadblocks to a better understanding of neural networks' internals is *polysemanticity*, where neurons appear to activate in multiple, semantically distinct contexts. Polysemanticity prevents us from identifying concise, human-understandable explanations for what neural networks are doing internally. One hypothesised cause of polysemanticity is *superposition*, where neural networks represent more features than they have neurons by assigning features to an overcomplete set of directions in activation space, rather than to individual neurons. Here, we attempt to identify those directions, using sparse autoencoders to reconstruct the internal activations of a language model. These autoencoders learn sets of sparsely activating features that are more interpretable and monosemantic than directions identified by alternative approaches, where interpretability is measured by automated methods. Moreover, we show that with our learned set of features, we can pinpoint the features that are causally responsible for counterfactual behaviour on the indirect object identification task (Wang et al., 2022) to a finer degree than previous decompositions. This work indicates that it is possible to resolve superposition in language models using a scalable, unsupervised method. Our method may serve as a foundation for future mechanistic interpretability work, which we hope will enable greater model transparency and steerability.

## 1 INTRODUCTION

Advances in artificial intelligence (AI) have resulted in the development of highly capable AI systems that make decisions for reasons we do not understand. This has caused concern that AI systems that we cannot trust are being widely deployed in the economy and in our lives, introducing a number of novel risks (Hendrycks et al., 2023), including potential future risks that AIs might deceive humans in order to accomplish undesirable goals (Ngo et al., 2022). Mechanistic interpretability seeks to mitigate such risks through understanding how neural networks calculate their outputs, allowing us to reverse engineer parts of their internal processes and make targeted changes to them (Cammarata et al., 2021; Wang et al., 2022; Elhage et al., 2021).

To reverse engineer a neural network, it is necessary to break it down into smaller units (features) that can be analysed in isolation. Using individual neurons as these units has had some success (Olah et al., 2020; Bills et al., 2023), but a key challenge has been that neurons are often *polysemantic*, activating for several unrelated types of feature (Olah et al., 2020). Also, for some types of network activations, such as the residual stream of a transformer, there is little reason to expect features to align with the neuron basis (Elhage et al., 2023).

Elhage et al. (2022b) investigate why polysemanticity might arise and hypothesise that it may result from models learning more distinct features than there are dimensions in the layer. They call this phenomenon *superposition*. Since a vector space can only have as many orthogonal vectors as it has dimensions, this means the network would learn an overcomplete basis of non-orthogonal features. Features must be sufficiently sparsely activating for superposition to arise because, without

---

[*]Equal contribution
Code to replicate experiments can be found at https://github.com/HoagyC/sparse_coding

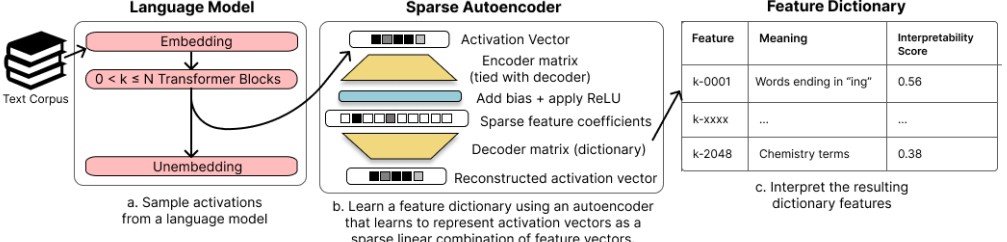

Figure 1: An overview of our method. We a) sample the internal activations of a language model, either the residual stream, MLP sublayer, or attention head sublayer; b) use these activations to train a neural network, a sparse autoencoder whose weights form a feature dictionary; c) interpret the resulting features with techniques such as OpenAI's autointerpretability scores (meanings here are illustrative only).

high sparsity, interference between non-orthogonal features prevents any performance gain from superposition. This suggests that we may be able to recover the network's features by finding a set of directions in activation space such that each activation vector can be reconstructed from a sparse linear combinations of these directions. This is equivalent to the well-known problem of sparse dictionary learning (Olshausen & Field, 1997).

Building on Sharkey et al. (2023), we train sparse autoencoders to learn these sets of directions. Our approach is also similar to Yun et al. (2021), who apply sparse dictionary learning to all residual stream layers in a language model simultaneously. Our method is summarised in Figure 1 and described in Section 2.

We then use several techniques to verify that our learned features represent a semantically meaningful decomposition of the activation space. First, we show that our features are on average more interpretable than neurons and other matrix decomposition techniques, as measured by autointerpretability scores (Section 3) (Bills et al., 2023). Next, we show that we are able to pinpoint the features used for a set task more precisely than other methods (Section 4). Finally, we run case studies on a small number of features, showing that they are not only monosemantic but also have predictable effects on the model outputs, and can be used for fine-grained circuit detection. (Section 5).

## 2    TAKING FEATURES OUT OF SUPERPOSITION WITH SPARSE DICTIONARY LEARNING

To take network features out of superposition, we employ techniques from *sparse dictionary learning* (Olshausen & Field, 1997; Lee et al., 2006). Suppose that each of a given set of vectors $\{\mathbf{x}_i\}_{i=1}^{n_{\text{vec}}} \subset \mathbb{R}^d$ is composed of a sparse linear combination of unknown vectors $\{\mathbf{g}_j\}_{j=1}^{n_{\text{gt}}} \subset \mathbb{R}^d$, i.e. $\mathbf{x}_i = \sum_j a_{i,j}\mathbf{g}_j$ where $\mathbf{a_i}$ is a sparse vector. In our case, the data vectors $\{\mathbf{x}_i\}_{i=1}^{n_{\text{vec}}}$ are internal activations of a language model, such as Pythia-70M (Biderman et al., 2023), and $\{\mathbf{g}_j\}_{j=1}^{n_{\text{gt}}}$ are unknown, ground truth network features. We would like learn a dictionary of vectors, called dictionary features, $\{\mathbf{f}_k\}_{k=1}^{n_{\text{feat}}} \subset \mathbb{R}^d$ where for any network feature $\mathbf{g}_j$ there exists a dictionary feature $\mathbf{f}_k$ such that $\mathbf{g}_j \approx \mathbf{f}_k$.

To learn the dictionary, we train an autoencoder with a sparsity penalty term on its hidden activations. The autoencoder is a neural network with a single hidden layer of size $d_{\text{hid}} = Rd_{\text{in}}$, where $d_{\text{in}}$ is the dimension of the language model internal activation vectors[1], and $R$ is a hyperparameter that controls the ratio of the feature dictionary size to the model dimension. We use the ReLU activation function in the hidden layer (Fukushima, 1975). We also use tied weights for our neural network, meaning the weight matrices of the encoder and decoder are transposes of each other.[2] Thus, on

---

[1] We mainly study residual streams in Pythia-70M and Pythia 410-M, for which the residual streams are of size $d_{\text{in}} = 512$ and $d_{\text{in}} = 1024$, respectively (Biderman et al., 2023)

[2] We use tied weights because (a) they encode our expectation that the directions which detect and define the feature should be the same or highly similar, (b) they halve the memory cost of the model, and (c) they remove

input vector $\mathbf{x} \in \{\mathbf{x}_i\}$, our network produces the output $\hat{\mathbf{x}}$, given by

$$\mathbf{c} = \text{ReLU}(M\mathbf{x} + \mathbf{b}) \tag{1}$$

$$\hat{\mathbf{x}} = M^T \mathbf{c} \tag{2}$$

$$= \sum_{i=0}^{d_{\text{hid}}-1} c_i \mathbf{f}_i \tag{3}$$

where $M \in \mathbb{R}^{d_{\text{hid}} \times d_{\text{in}}}$ and $\mathbf{b} \in \mathbb{R}^{d_{\text{hid}}}$ are our learned parameters, and $M$ is normalised row-wise[3]. Our parameter matrix $M$ is our feature dictionary, consisting of $d_{\text{hid}}$ rows of dictionary features $\mathbf{f}_i$. The output $\hat{\mathbf{x}}$ is meant to be a reconstruction of the original vector $\mathbf{x}$, and the hidden layer $\mathbf{c}$ consists of the coefficients we use in our reconstruction of $\mathbf{x}$.

Our autoencoder is trained to minimise the loss function

$$\mathcal{L}(\mathbf{x}) = \underbrace{\frac{||\mathbf{x} - \hat{\mathbf{x}}||_2^2}{\dim(\mathbf{x})}}_{\text{Reconstruction loss}} + \underbrace{\alpha ||\mathbf{c}||_1}_{\text{Sparsity loss}} \tag{4}$$

where $\alpha$ is a hyperparameter controlling the sparsity of the reconstruction, $l_m$ is the width of the original activation. The $\ell^1$ loss term on $\mathbf{c}$ encourages our reconstruction to be a sparse linear combination of the dictionary features. It can be shown empirically (Sharkey et al., 2023) and theoretically (Wright & Ma, 2022) that reconstruction with an $\ell^1$ penalty can recover the ground-truth features that generated the data.

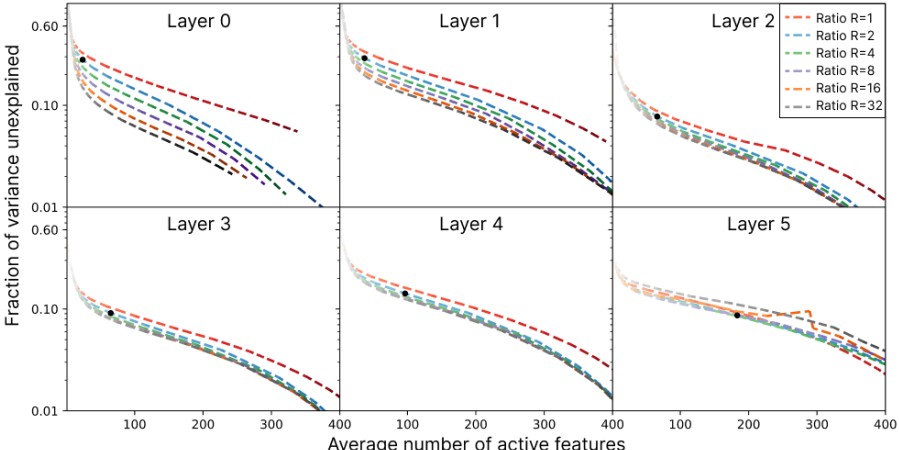

Figure 2: The tradeoff between the average number of active features and the fraction of the variance that is unexplained, as the $\ell^1$ coefficient $\alpha$ is varied. Model is Pythia70M. Black dot represents the $R = 2$, $\alpha = 0.00086$ point used for autointerpretation.

---

ambiguity about whether the learned direction should be interpreted as the encoder or decoder direction. They do not reduce performance when training on residual stream data but we have observed some reductions in performance when using MLP data.

[3]Normalisation of the rows (dictionary features) prevents the model from reducing the sparsity loss term $||\mathbf{c}||_1$ by increasing the size of the feature vectors in $M$.

| Feature | Description (Generated by GPT-4) | Interpretability Score |
|---------|-----------------------------------|------------------------|
| 1-0000 | parts of individual names, especially last names. | 0.33 |
| 1-0001 | actions performed by a subject or object. | -0.11 |
| 1-0002 | instances of the letter 'W' and words beginning with 'w'. | 0.55 |
| 1-0003 | the number '5' and also records moderate to low activation for personal names and some nouns. | 0.57 |
| 1-0004 | legal terms and court case references. | 0.19 |

Table 1: Results of autointerpretation on the first five features found in the layer 1 residual stream, with $R = 2$, $\alpha = 0.00086$ on Pythia70m. Autointerpretation produces a description of what the feature means and a score for how well that description predicts other activations.

## 3 INTERPRETING DICTIONARY FEATURES

### 3.1 INTERPRETABILITY AT SCALE

Having learned a set of dictionary features, we want to understand whether our learned features display reduced polysemanticity, and are therefore more interpretable. To do this in a scalable manner, we require a metric to measure how interpretable a dictionary feature is. We use the automated approach introduced in Bills et al. (2023) because it scales well to measuring interpretability on the thousands of dictionary features our autoencoders learn. In summary, the autointerpretability procedure takes samples of text where the dictionary feature activates, asks a language model to write a human-readable interpretation of the dictionary feature, and then prompts the language model to use this description to predict the dictionary feature's activation on other samples of text. The correlation between the model's predicted activations and the actual activations is that feature's interpretability score. See Appendix A and Bills et al. (2023) for further details.

We show descriptions and top-and-random scores for five dictionary features from the layer 1 residual stream in Table 1. The features shown are the first five under the (arbitrary) ordering in the dictionary.

### 3.2 SPARSE DICTIONARY FEATURES ARE MORE INTERPRETABLE THAN BASELINES

We assess our interpretability scores against a variety of alternative methods for finding dictionaries of features in language models. In particular, we compare interpretability scores on our dictionary features to those produced by a) the default basis, b) random directions, c) Principal Component Analysis (PCA), and d) Independent Component Analysis (ICA). For the random directions and for the default basis in the residual stream, we replace negative activations with zeros so that all feature activations are nonnegative [4].

Figure 3 shows that our dictionary features are far more interpretable by this measure than dictionary features found by comparable techniques. We find that the strength of this effect declines as we move through the model, being comparable to ICA in layer 4 and showing minimal improvement in the final layer.

This could be a result of our use of a consistent $\alpha = 0.00086$, $R = 2$ in our automatic interpretation results, which as seen in Figure 2 led to a higher number of average active features in the later layers. However, it may also indicate that sparse autoencoders work less well in later layers but also may be connected to the difficulties of automatic interpretation, both because by building on earlier layers, later features may be more complex, and because they are often best explained by their effect on the output.Bills et al. (2023) showed that GPT-4 is able to generate explanations that are very close to the average quality of the human-generated explanations given similar data. However, they also showed that current LLMs are limited in the kinds of patterns that they can find, sometimes struggling to find patterns that center around next or previous tokens rather than the current token, and in the current protocol are unable to verify outputs by looking at changes in output or other data.

---

[4]For PCA we use an online estimation approach and run the decomposition on the same quantity of data we used for training the autoencoders. For ICA, due to the slower convergence times, we run on only 2GB of data, approximately 4 million activations for the residual stream and 1m activations for the MLPs.

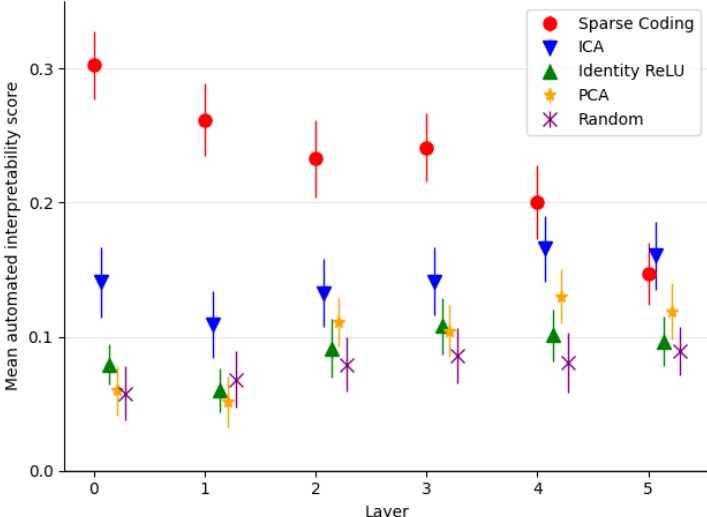

Figure 3: Average top-and-random autointerpretability score of our learned directions in the residual stream, compared to a number of baselines, using 150 features each. Error bars show 95% confidence intervals around means. The feature dictionaries used here were trained for 10 epochs using $\alpha = .00086$ and $R = 2$ on Pythia 70M.

We do show, in Section 5, a method to see a feature's causal effect on the output logits by hand, but we currently do not send this information to the language model for hypothesis generation. The case studies section also demonstrates a closing parenthesis dictionary feature, showing that these final layer features can give insight into the model's workings.

See Appendix C for a fuller exploration of different learned dictionaries through the lens of automatic interpretability, looking at both the MLPs and the residual stream.

## 4 IDENTIFYING CAUSALLY-IMPORTANT DICTIONARY FEATURES FOR INDIRECT OBJECT IDENTIFICATION

In this section, we quantify whether our learned dictionary features localise a specific model behaviour more tightly than the PCA decomposition of the model's activations. We do this via activation patching, a form of causal mediation analysis (Vig et al., 2020), through which we edit the model's internal activations along the directions indicated by our dictionary features and measure the changes to the model's outputs. We find that our dictionary features require fewer patches to reach a given level of KL divergence on the task studied than comparable decompositions (Figure 4).

Specifically, we study model behaviour on the Indirect Object Identification (IOI) task (Wang et al., 2022), in which the model completes sentences like "Then, Alice and Bob went to the store. Alice gave a snack to ___". This task was chosen because it captures a simple, previously-studied model behaviour, which in particular has been widely explored through causal mediation analysis (Wang et al., 2022) (Conmy et al., 2023) and it captures a simple model behaviour. Recall that the training of our feature dictionaries does not emphasize any particular task.

### 4.1 ADAPTING ACTIVATION PATCHING TO DICTIONARY FEATURES

In our experiment, we run the model on a counterfactual target sentence, which is a variant of the base IOI sentence with the indirect object changed (e.g., with "Bob" replaced by "Vanessa"); save the encoded activations of our dictionary features; and use the saved activations to edit the model's residual stream when run on the base sentence.

In particular, we perform the following procedure. Fix a layer of the model to intervene on. Run the model on the target sentence, saving the model output logits $\mathbf{y}$ and the encoded features $\bar{\mathbf{c}}_1, ..., \bar{\mathbf{c}}_k$ of that layer at each of the $k$ tokens. Then, run the model on the base sentence up through the intervention layer, compute the encoded features $\mathbf{c}_1, ..., \mathbf{c}_k$ at each token, and at each position replace the residual stream vector $\mathbf{x}_i$ with the patched vector

$$\mathbf{x}_i' = \mathbf{x}_i + \sum_{j \in F} (\bar{\mathbf{c}}_{i,j} - \mathbf{c}_{i,j}) \mathbf{f}_j$$

where $F$ is the subset of the features which we intervene on (we describe the selection process for $F$ later in this section). Let $\mathbf{z}$ denote the output logits of the model when you finish applying it to the patched residual stream $\mathbf{x}_1', ..., \mathbf{x}_k'$. Finally, compute the KL divergence $D_{KL}(\mathbf{z}||\mathbf{y})$, which measures how close the patched model's predictions are to the target's. We compare these interventions to equivalent interventions using principal components found as in Section 3.2.

To select the feature subset $F$, we use the Automated Circuit Discovery (ACDC) algorithm of Conmy et al. (2023). In particular, we use their Algorithm 4.1 on our features, treating them as a flat computational graph in which every feature contributes an independent change to the $D_{KL}$ output metric, as described above and averaged over a test set of 50 IOI data points. The result is an ordering on the features so that patching the next feature usually results in a smaller $D_{KL}$ loss than each previous feature. Then our feature subsets $F$ are the first $k$ features under this ordering. We applied ACDC separately on each decomposition.

## 4.2 PRECISE LOCALISATION OF IOI DICTIONARY FEATURES

We show in Figure 4 that our sparse feature dictionaries allow the same amount of model editing, as measured by KL divergence from the target, in fewer patches (Left) and with smaller edit magnitude (Right) than the PCA decomposition. We also show that this does not happen if we train a non-sparse dictionary ($\alpha = 0$). However, dictionaries with a larger sparsity coefficient $\alpha$ have lower overall reconstruction accuracy which appears in Figure 4 as a larger minimum KL divergence. In Figure 4 we consider interventions on layer 11 of the residual stream, and we plot interventions on other layers in Appendix F.

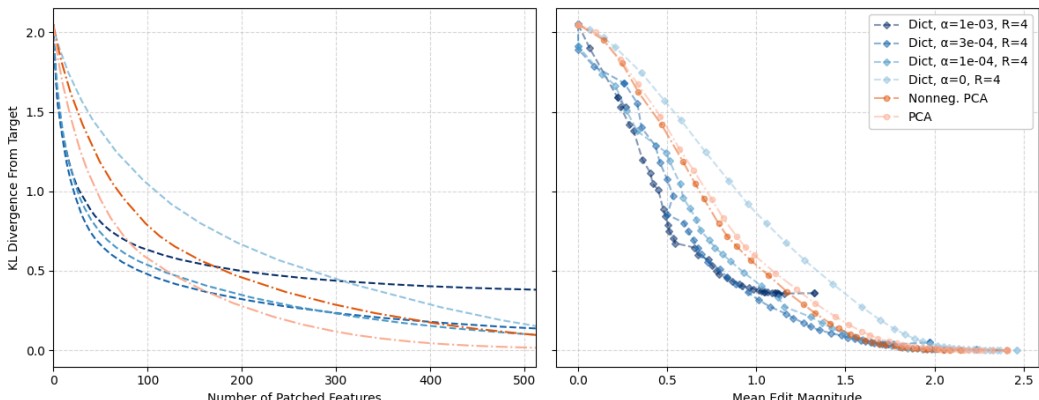

Figure 4: (Left) Number of features patched vs KL divergence from target, using various residual stream decompositions. We find that patching a relatively small number of dictionary features is more effective than patching PCA components and features from the non-sparse $\alpha = 0$ dictionary. (Right) Mean edit magnitude vs KL divergence from target as we increase the number of patched features. We find that our sparse dictionaries improve the Pareto frontier of edit magnitude vs thoroughness of editing. In both figures, the feature dictionaries were trained on the first 10,000 elements of the Pile (Gao et al., 2020) (approximately 7 million activations) using the indicated $\alpha$ values and $R = 4$, on layer 11 of Pythia-410M (see Appendix F for results on other layers).

# 5 CASE STUDIES

In this section, we investigate individual dictionary features, highlighting several that appear to correspond to a single human-understandable explanation (i.e., that are monosemantic). We perform three analyses of our dictionary features to determine their semantic meanings: (1) *Input*: We identify which tokens activate the dictionary feature and in which contexts, (2) *Output*: We determine how ablating the feature changes the output logits of the model, and (3) *Intermediate features*: We identify the dictionary features in previous layers that cause the analysed feature to activate.

## 5.1 INPUT: DICTIONARY FEATURES ARE HIGHLY MONOSEMANTIC

We first analyse our dictionary directions by checking what text causes them to activate. An idealised monosemantic dictionary feature will only activate on text corresponding to a single human-understandable concept, whereas a polysemantic dictionary feature might activate in unrelated contexts.

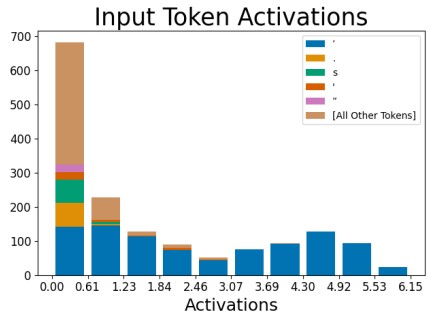
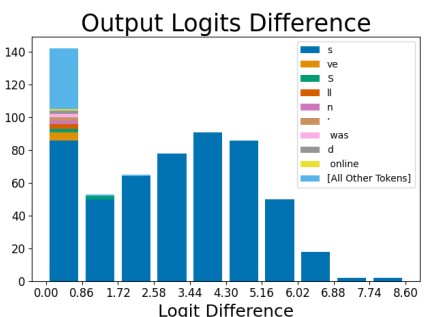

Figure 5: Histogram of token counts for dictionary feature 556 in layer 4 of Pythia-70M-deduped. (Left) For all datapoints that activate the feature, we show the count of each token in each activation range. The majority of activations are apostrophes, particularly for higher activations. Notably the lower activating tokens are conceptually similar to apostrophes, such as other punctuation. (Right) We show which token predictions are suppressed by ablating the feature, as measured by the difference in logits between the ablated and unablated model. We find that the token whose prediction decreases the most is the "s" token. Note that there are 12k logits negatively effected, but we set a threshold of 0.1 for visual clarity. The autoencoder hyperparameters used were $R = 4, \alpha = 0.0014$.

To better illustrate the monosemanticity of certain dictionary features, we plot the histogram of activations across token activations. This technique only works for dictionary features that activate for a small set of tokens. We find dictionary features that only activate on apostrophes (Figure 5); periods; the token " the"; and newline characters. The apostrophe feature in Figure 5 stands in contrast to the default basis for the residual stream, where the dimension that most represents an apostrophe is displayed in Figure 10 in Appendix D.1; this dimension is polysemantic since it represents different information at different activation ranges.

Although the dictionary feature discussed in the previous section activates only for apostrophes, it does not activate on *all* apostrophes. This can be seen in Figures 13 and 14 in Appendix D.2, showing two other apostrophe-activating dictionary features, but for different contexts (such as "[I/We/They]'ll" and "[don/won/wouldn]'t"). Details for how we searched and selected for dictionary features can be found in Appendix D.3.

## 5.2 OUTPUT: DICTIONARY FEATURES HAVE INTUITIVE EFFECTS ON THE LOGITS

In addition to looking at which tokens activate the dictionary feature, we investigate how dictionary features affect the model's output predictions for the next token by ablating the feature from the residual stream[5]. If our dictionary feature is interpretable, subtracting its value from the residual

---

[5]Specifically we use less-than-rank-one ablation, where we lower the activation vector in the direction of the feature only up to the point where the feature is no longer active.

stream should have a logical effect on the predictions of the next token. We see in Figure 5 (Right) that the effect of removing the apostrophe feature mainly reduces the logit for the following "s". This matches what one would expect from a dictionary feature that detects apostrophes and is used by the model to predict the "s" token that would appear immediately after the apostrophe in possessives and contractions like "let's".

### 5.3 Intermediate Features: Dictionary Features Allow Automatic Circuit Detection

We can also understand dictionary features in relation to the upstream and downstream dictionary features: given a dictionary feature, which dictionary features in previous layers cause it to activate, and which dictionary features in later layers does it cause to activate?

To automatically detect the relevant dictionary features, we choose a target dictionary feature such as layer 5's feature for tokens in parentheses which predicts a closing parentheses (Figure 6). For this target dictionary feature, we find its maximum activation $M$ across our dataset, then sample 20 contexts that cause the target feature to activate in the range $[M/2, M]$. For each dictionary feature in the previous layer, we rerun the model while ablating this feature and sort the previous-layer features by how much their ablation decreased the target feature. If desired, we can then recursively apply this technique to the dictionary features in the previous layer with a large impact. The results of this process form a causal tree, such as Figure 6.

Being the last layer, layer 5's role is to output directions that directly correspond to tokens in the unembedding matrix. In fact, when we unembed feature $5_{2027}$, the top-tokens are all closing parentheses variations. Intuitively, previous layers will detect all situations that precede closing parentheses, such as dates, acronyms, and phrases.

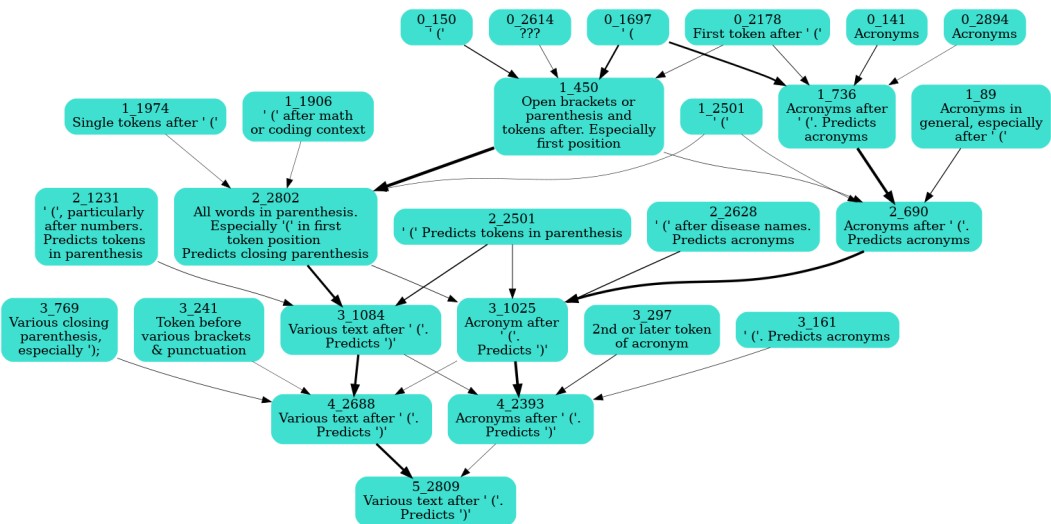

Figure 6: Circuit for the closing parenthesis dictionary feature, with human interpretations of each feature shown. Edge thickness indicates the strength of the causal effect between dictionary features in successive residual stream layers, as measured by ablations. Many dictionary features across layers have similar interpretations and often point in similar directions in activation space, as measured by cosine similarity. Model used was Pythia-70M-deduped, with the autoencoder hyperparameters $R = 4, \alpha = 0.0014$.

## 6 Discussion

### 6.1 Related Work

Several previous works have attempted to decompose language representations into sparsely-activating features, varying both the representation studied and the technique used. Our approach,

training a neural network with a sparsity term in the loss function, is similar to the approaches in Faruqui et al. (2015); Subramanian et al. (2018); Sharkey et al. (2023). In other works, such as Yun et al. (2021); Zhang et al. (2019), the decomposition is found via the FISTA algorithm, and Murphy et al. (2012) uses the Non-Negative Sparse Embeddings method. Of these works, Faruqui et al. (2015); Subramanian et al. (2018); Zhang et al. (2019); Murphy et al. (2012) applied these techniques to word embeddings, while only Sharkey et al. (2023); Yun et al. (2021) found sparse decompositions of the activations of a language model. Many of these works, including Murphy et al. (2012); Subramanian et al. (2018); Yun et al. (2021) also find improved interpretability of their features, as measured by techniques such as crowd-sourced judgements, the word intrusion detection test, and word-level polysemy disambiguation, respectively.

The works most similar to ours are Sharkey et al. (2023), which inspired this work, and Subramanian et al. (2018). The latter use sparse autoencoders to learn their decomposition of word embeddings, though for their main results they use losses which train the learned features to approximate a sparse binary unit, finding in preliminary experiments that this outperformed the use of an $\ell^1$ penalty.

Other previous works have tried to encourage sparsity in neural networks via changes to the architecture or training process. These approaches include altering the attention mechanism (Correia et al., 2019), adding $\ell^1$ penalties to neuron activations (Kasioumis et al., 2021; Georgiadis, 2019), pruning neurons (Frankle & Carbin, 2018), and using the softmax function as the non-linearity in the MLP layers (Elhage et al., 2022a). However, training a state-of-the-art foundation model with these additional constraints is difficult (Elhage et al., 2022a), and improvements to interpretability are not always realized (Meister et al., 2021).

## 6.2 LIMITATIONS AND FUTURE WORK

The approach we present in this paper found interpretable directions, but depending on the choice of hyperparameters leaves a significant fraction of the model's variance unexplained (Figure 2). Future work could seek to improve the Pareto frontier of sparsity and reconstruction accuracy by exploring alternative architectures for the autoencoder or incorporating information about the weights of the model or dictionary features found in adjacent layers into the training process. This approach could also be applied to other components of a transformer, such as the output of the MLP or attention sublayers, as our attempt to find sparse directions in the MLP layer met only mixed success (see Appendix C).

In Section 4, we show that for the IOI task, behaviour is dependent on a relatively small number of features. We expect that, because our dictionary is trained in a task-agnostic way, these result will generalize to similar tasks and behaviours, but more work is needed to confirm this suspicion. If this property generalizes, we would have a set of features which allow for understanding many model behaviours using just a few features per behaviour. We would also like to trace the causal dependencies between features in different layers, with the overarching goal of providing a lens for viewing language models under which causal dependencies are sparse. This would hopefully be a step towards the eventual goal of building an end-to-end understanding of how a model computes its outputs.

## 6.3 CONCLUSION

Sparse autoencoders are a scalable, unsupervised approach to disentangling language model network features from superposition. Our approach requires only unlabelled model activations and uses orders of magnitude less compute than the training of the original models. We have demonstrated that the dictionary features we learn are more interpretable by autointerpretation, letting us pinpoint the features responsible for a given behaviour more finely, and are more monosemantic than comparable methods. This approach could facilitate the mapping of model circuits, targeted model editing, and a better understanding of model representations.

An ambitious dream in the field of interpretability is enumerative safety (Elhage et al., 2022b): producing a human-understandable explanation of a model's computations in terms of a complete list of the model's features and thereby providing a guarantee that the model will not perform dangerous behaviours such as deception. We hope that the techniques we presented in this paper also provide a step towards achieving this ambition.

ACKNOWLEDGMENTS

We would like to thank the OpenAI Researcher Access Program for their grant of model credits for the autointerpretation and CoreWeave for providing EleutherAI with the computing resources for this project. We also thank Nora Belrose, Arthur Conmy, Jake Mendel, and the OpenAI Automated Interpretability Team (Jeff Wu, William Saunders, Steven Bills, Henk Tillman, and Daniel Mossing) for valuable discussions regarding the design of various experiments. We thank Wes Gurnee, Adam Jermyn, Stella Biderman, Leo Gao, Curtis Huebner, Scott Emmons, and William Saunders for their feedback on earlier versions of this paper. Thanks to Delta Hessler for proofreading. AE and LR are supported by the Long Term Future Fund. RH is supported by an Open Philanthropy grant. HC was greatly helped by the MATS program, funded by AI Safety Support.

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

## A    AUTOINTERPRETATION PROTOCOL

The autointerpretability process consists of five steps and yields both an interpretation and an autointerpretability score:

1. On each of the first 50,000 lines of OpenWebText, take a 64-token sentence-fragment, and measure the feature's activation on each token of this fragment. Feature activations are rescaled to integer values between 0 and 10.

2. Take the 20 fragments with the top activation scores and pass 5 of these to GPT-4, along with the rescaled per-token activations. Instruct GPT-4 to suggest an explanation for when the feature (or neuron) fires, resulting in an interpretation.

3. Use GPT-3.5[6] to simulate the feature across another 5 highly activating fragments and 5 randomly selected fragments (with non-zero variation) by asking it to provide the per-token activations.

4. Compute the correlation of the simulated activations and the actual activations. This correlation is the autointerpretability score of the feature. The texts chosen for scoring a feature can be random text fragments, fragments chosen for containing a particularly high activation of that feature, or an even mixture of the two. We use a mixture of the two unless otherwise noted, also called 'top-random' scoring.

5. If, amongst the 50,000 fragments, there are fewer than 20 which contain non-zero variation in activation, then the feature is skipped entirely.

Although the use of random fragments in Step 4 is ultimately preferable given a large enough sample size, the small sample sizes of a total of 640 tokens used for analysis mean that a random sample will likely not contain any highly activating examples for all but the most common features, making top-random scoring a desirable alternative.

## B    SPARSE AUTOENCODER TRAINING

To train the sparse autoencoder described in Section 2, we use data from the Pile (Gao et al., 2020), a large, public webtext corpus. We run the model that we want to interpret over this text while caching and saving the activations at a particular layer. These activations then form a dataset, which we use to train the autoencoders. The autoencoders are trained with the Adam optimiser with a learning rate of 1e-3 and are trained on 5-50M activation vectors for 1-3 epochs, with larger dictionaries taking longer to converge. A single training run using this quantity of data completes in under an hour on a single A40 GPU.

When varying the hyperparameter $\alpha$ which controls the importance of the sparsity loss term, we consistently find a smooth tradeoff between the sparsity and accuracy of our autoencoder, as shown in Figure 2. The lack of a 'bump' or 'knee' in these plots provides some evidence that there is not a single correct way to decompose activation spaces into a sparse basis, though to confirm this would require many additional experiments.

---

[6] While the process described in Bills et al. (2023) uses GPT-4 for the simulation step, we use GPT-3.5. This is because the simulation protocol requires the model's logprobs for scoring, and OpenAI's public API for GPT-3.5 (but not GPT-4) supports returning logprobs.

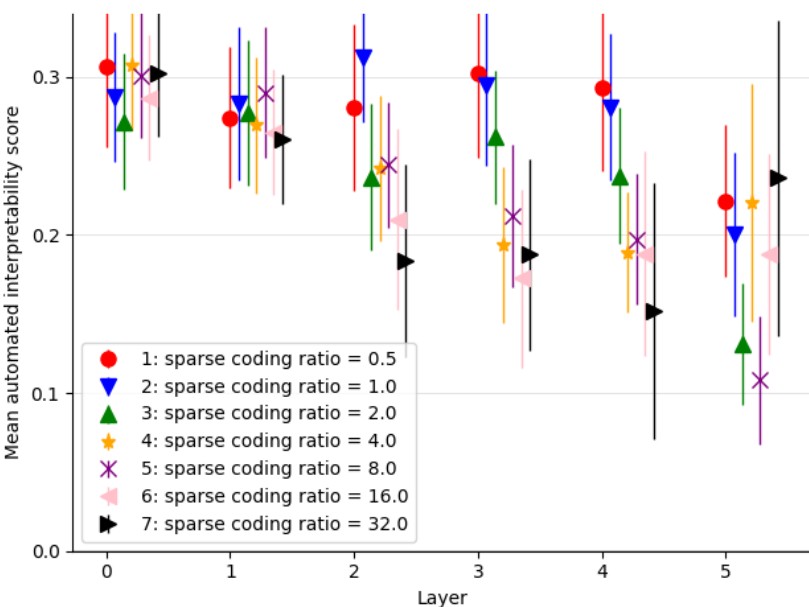

Figure 7: Comparison of average interpretability scores across dictionary sizes. All dictionaries were trained on 20M activation vectors obtained by running Pythia-70M over the Pile with $\alpha = .00086$.

## C    FURTHER AUTOINTERPRETATION RESULTS

### C.1    INTERPRETABILITY IS CONSISTENT ACROSS DICTIONARY SIZES

We find that larger interpretability scores of our feature dictionaries are not limited to overcomplete dictionaries (where the ratio, $R$, of dictionary features to model dimensions is $> 1$), but occurs even in dictionaries that are smaller than the underlying basis, as shown in Figure 7. These small dictionaries are able to reconstruct the activation vectors less accurately, so with each feature being similarly interpretable, the larger dictionaries will be able to explain more of the overall variance.

### C.2    HIGH INTERPRETABILITY SCORES ARE NOT AN ARTEFACT OF TOP SCORING

A possible concern is that the autointerpretability method described in Section 3 combines top activating fragments (which are usually large) with random activations (which are usually small), making it relatively easy to identify activations. Following the lead of Bills et al. (2023), we control for this by recomputing the autointerpretation scores by modifying Step 3 using only randomly selected fragments. With large sample sizes, using random fragments should be the true test of our ability to interpret a potential feature. However, the features we are considering are heavy-tailed, so with limited sample sizes, we should expect random samples to underestimate the true correlation.

In Figure 8 we show autointerpretability scores for fragments using only random fragments. Matching Bills et al. (2023), we find that random-only scores are significantly smaller than top-and-random scores, but also that our learned features still consistently outperform the baselines, especially in the early layers. Since our learned features are more sparse than the baselines and thus, activate less on a given fragment, this is likely to underestimate the performance of sparse coding relative to baselines.

An additional potential concern is that the structure of the autoencoders allows them to be sensitive to less than a full direction in the activation space, resulting in an unfair comparison. We show in Appendix G that this is not the source of the improved performance of sparse coding.

While the residual stream can usually be treated as a vector space with no privileged basis (a basis in which we would expect changes to be unusually meaningful, such as the standard basis after a non-linearity in an MLP), it has been noted that there is a tendency for transformers to store information in the residual stream basis(Dettmers et al., 2022), which is believed to be caused by the Adam

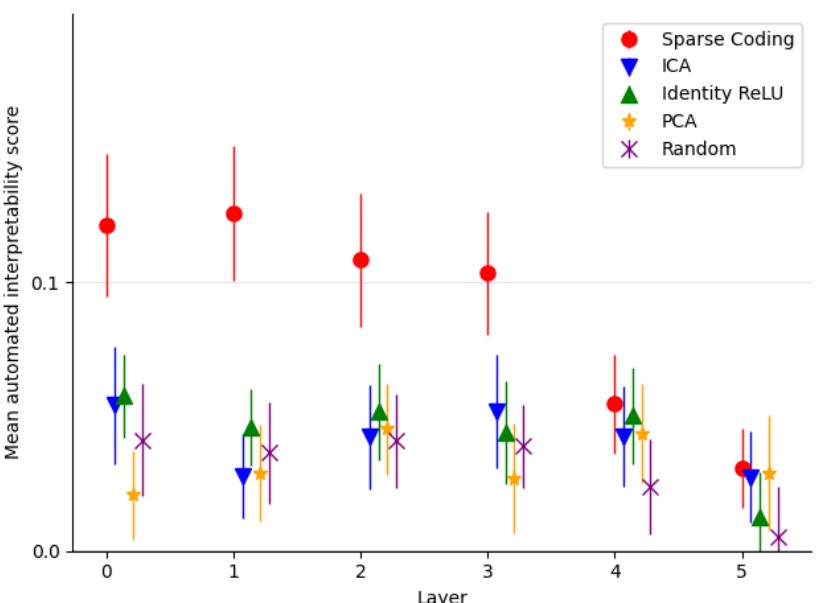

Figure 8: Random-only interpretability scores across each layer, a measure of how well the interpretation of the top activating cluster is able to explain the entire range of activations, $\alpha = 0.00086$, $R = 2$, Pythia70M.

optimiser saving gradients with finite precision in the residual basis(Elhage et al., 2023). We do not find residual stream basis directions to be any more interpretable than random directions.

### C.3 INTERPRETING THE MLP SUBLAYER

Our approach of learning a feature dictionary and interpreting the resulting features can, in principle, be applied to any set of internal activations of a language model, not just the residual stream. Applying our approach to the MLP sublayer of a transformer resulted in mixed success. Our approach still finds many features that are more interpretable than the neurons. However, our architecture also learns many dead features, which never activate across the entire corpus. In some cases, there are so many dead features that the set of living features does not form an overcomplete basis. For example, in a dictionary with twice as many features as neurons, less than half might be active enough to perform automatic interpretability. The exceptions to this are the early layers, where a large fraction of them are active.

For learning features in MLP layers, we find that we retain a larger number of features if we use a different matrix for the encoder and decoder, so that Equations 1 and 2 become

$$
\begin{aligned}
\mathbf{c} &= ReLU(M_e\mathbf{x} + \mathbf{b}) & (5) \\
\hat{\mathbf{x}} &= M_d^T\mathbf{c} & (6)
\end{aligned}
$$

We are currently working on methods to overcome this and find truly overcomplete bases in the middle and later MLP layers.

### C.4 INTERPRETABILITY SCORES CORRELATE WITH KURTOSIS AND SKEW OF ACTIVATION

It has been shown that the search for sparse, overcomplete dictionaries can be reformulated in terms of the search for directions that maximise the $\ell^4$-norm (Qu et al., 2019).

We offer a test of the utility of this by analysing the correlation between interpretability and a number of properties of learned directions. We find that there is a correlation of 0.19 and 0.24 between the

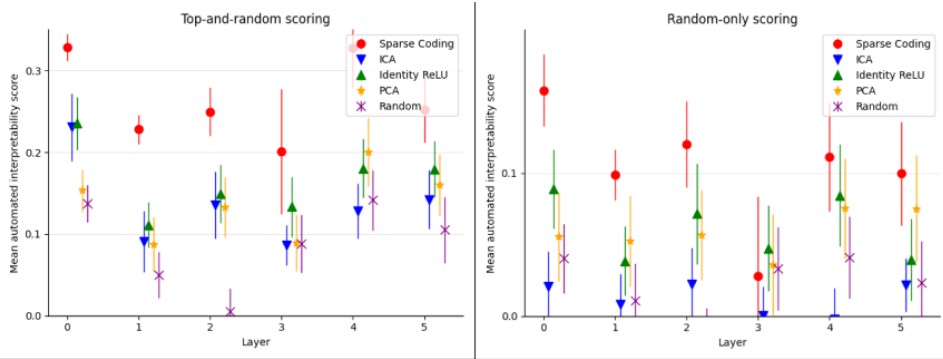

Figure 9: Top-and-random and random-only interpretability scores for across each MLP layer, using an $\ell^1$ coefficient $\alpha = 3.2e - 4$ and dictionary size ratio $R = 1$, Pythia70M.

degree of positive skew and kurtosis respectively that feature activations have and their top-and-random interpretability scores, as shown in Table 2.

| Moment | Correlation with top-random interpretability score |
|---|---|
| Mean | -0.09 |
| Variance | 0.02 |
| Skew | 0.20 |
| Kurtosis | 0.15 |

Table 2: Correlation of interpretability score with feature moments across residual stream results, all layers of Pythia70M, with dictionary size ratios $R \in \{0.5, 1, 2, 4, 8\}$.

This also accords with the intuitive explanation that the degree of interference due to other active features will be roughly normally distributed by the central limit theorem. If this is the case, then features will be notable for their heavy-tailedness.

This also explains why Independent Component Analysis (ICA), which maximises the non-Gaussianity of the found components, is the best performing of the alternatives that we considered.

## D   QUALITATIVE FEATURE ANALYSIS

### D.1   RESIDUAL STREAM BASIS

Figure 10 gives a token activation histogram of the residual stream basis. Connecting this residual stream dimension to the apostrophe feature from Figure 5, this residual dimension was the 10th highest dimension read from the residual stream by our feature[7].

### D.2   EXAMPLES OF LEARNED FEATURES

Other features from Pythia-70M-deduped (layer 4, $R = 4, \alpha = 0.0014$) are shown in Figures 11, 12, 13, and 14.

### D.3   FEATURE SEARCH DETAILS

We searched for the apostrophe feature using the sentence " I don't know about that. It is now up to Dave"', and seeing which feature (or residual stream dimension) activates the most for the last apostrophe token. The top activating feature in our dictionary was an outlier dimension feature (i.e., a feature direction that mainly reads from an outlier dimension of the residual stream), the apostrophes after O (and predicted O'Brien, O'Donnell, O'Connor, O'clock, etc), then the apostrophe-preceding-s feature.

---

[7]The first 9 did not have apostrophes in their top-activations like dimension 21.

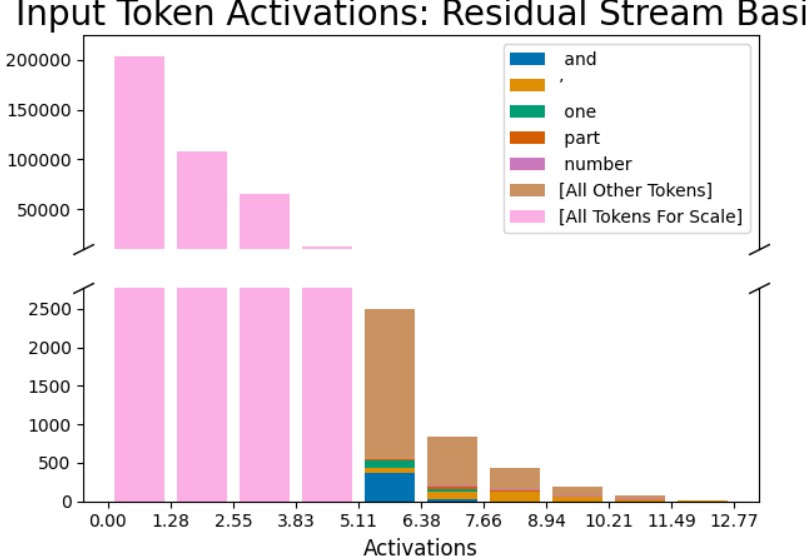

Figure 10: Histogram of token counts in the neuron basis (dimension 21 of layer 4 of Pythia-70M-deduped, positive direction). Although there are a large fraction of apostrophes in the upper activation range, this only explains a very small fraction of the variance for middle-to-lower activation ranges.

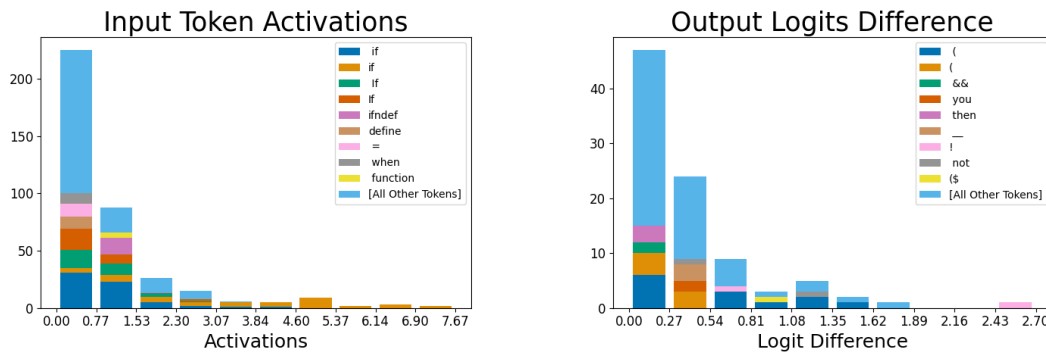

Figure 11: 'If' feature in coding contexts (Feature #1087 from Pythia-70M-deduped layer 4, with autoencoder hyperparameters $R = 4, \alpha = 0.0014$).

For the residual basis dimension, we searched for max and min activating dimensions (since the residual stream can be both positive and negative), where the top two most positive dimensions were outlier dimensions, the top two negative dimensions were our displayed one and another outlier dimension, respectively.

## D.4    FAILED INTERPRETABILITY METHODS

We attempted a weight-based method going from the dictionary in layer 4 to the dictionary in layer 5 by multiplying a feature by the MLP and checking the cosine similarity with features in layer 5. There were no meaningful connections. Additionally, it's unclear how to apply this to the Attention sublayer since we'd need to see which position dimension the feature is in. We expected this failed by going out of distribution.

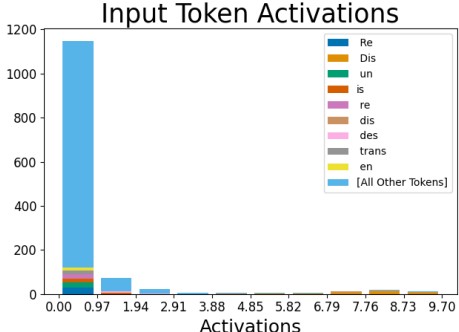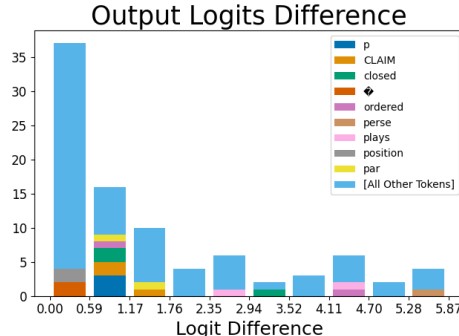

Figure 12: 'Dis' token-level feature showing bigrams, such as 'disCLAIM', 'disclosed', 'disordered', etc. (Feature #1128 from Pythia-70M-deduped layer 4, with autoencoder hyperparameters $R = 4, \alpha = 0.0014$).

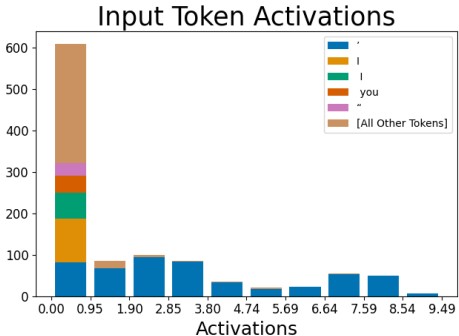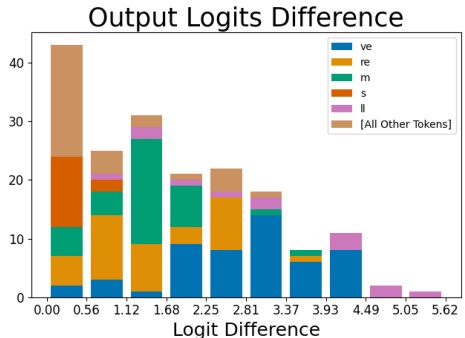

Figure 13: Apostrophe feature in "I'll"-like contexts (Feature #1243 from Pythia-70M-deduped layer 4, with autoencoder hyperparameters $R = 4, \alpha = 0.0014$).

## E   NUMBER OF ACTIVE FEATURES

In Figure 15 we see that, for residual streams, we consistently learn dictionaries that are at least 4x overcomplete before some features start to drop out completely, with the correct hyperparameters. For MLP layers you see large numbers of dead features even with hyperparameter $\alpha = 0$. These figures informed the selection of $\alpha = 8.6e - 4$ and $\alpha = 3.2e - 4$ that went into the graphs in Section 3 for the residual stream and MLP respectively. Due to the large part of the input space that is never used due to the non-linearity, it is much easier for MLP dictionary features to become stuck at a position where they hardly ever activate. In future we plan to reinitialise such 'dead features' to ensure that we learn as many useful dictionary features as possible.

## F   EDITING IOI BEHAVIOUR ON OTHER LAYERS

In Figure 16 we show results of the procedure in Section 4 across a range of layers in Pythia-410M.

## G   TOP K COMPARISONS

As mentioned in Section 3, the comparison directions learnt by sparse coding and those in the baselines are not perfectly even. This is because, for example, a PCA direction is active to an entire half-space on one side of a hyperplane through the origin, whereas a sparse coding feature activates on less than a full direction, being only on the far side of a hyperplane that does not intersect the origin. This is due to the bias applied before the activation, which is, in practice, always negative. To test whether this difference is responsible for the higher scores, we run a variant of PCA and ICA in which we have a fixed number of directions, K, which can be active for any single datapoint. We

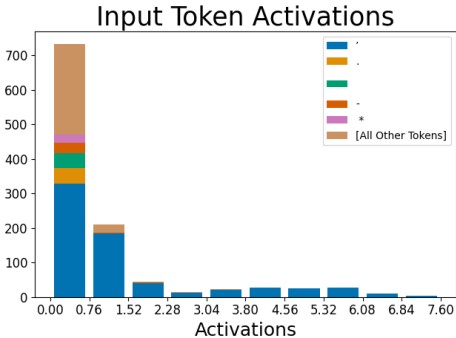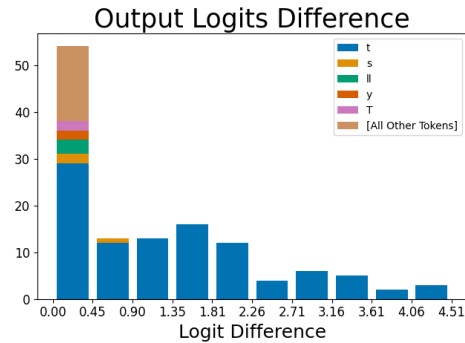

Figure 14: Apostrophe feature in "don't"-like contexts. (Feature #101 from Pythia-70M-deduped layer 4, with autoencoder hyperparameters $R = 4, \alpha = 0.0014$).

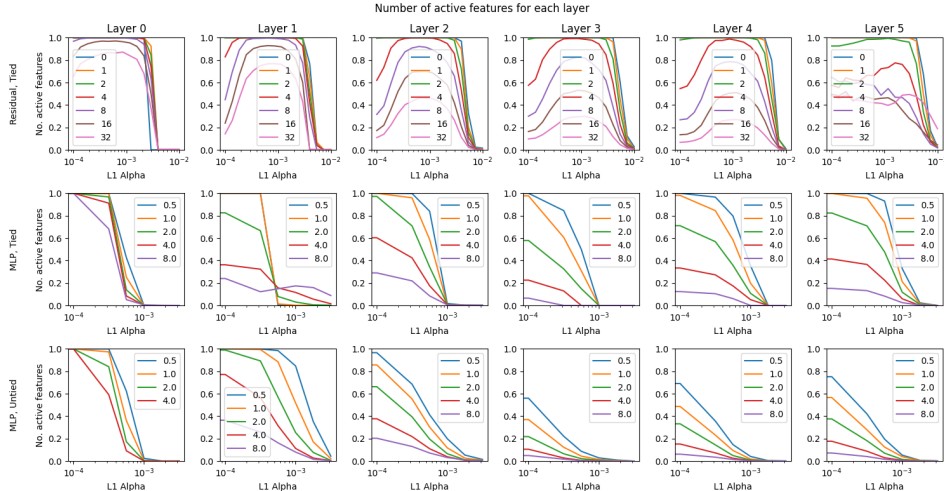

Figure 15: The number of features that are active, defined as activating more than 10 times across 10M datapoints, changes with sparsity hyperparamter $\alpha$ and dictionary size ratio $R$.

set this K to be equal to the average number of active features for a sparse coding dictionary with ratio $R = 1$ and $\alpha = 8.6e - 4$ trained on the layer in question. We compare the results in Figure 17, showing that this change does not explain more than a small fraction of the improvement in scores.

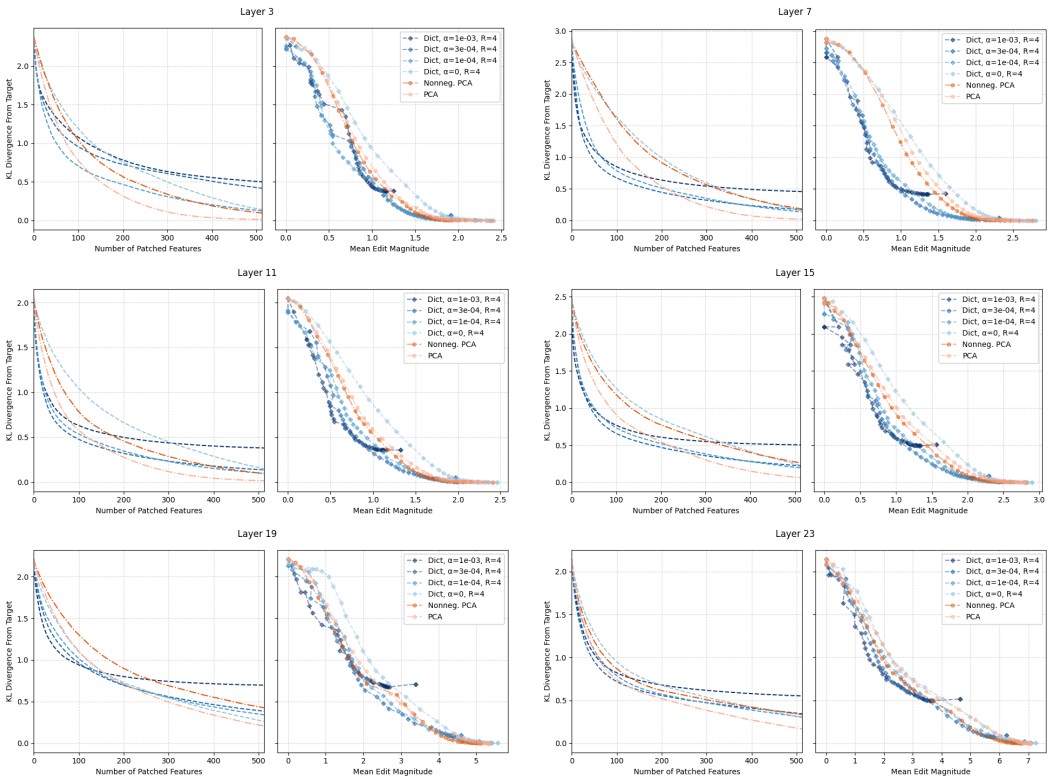

Figure 16: Divergence from target output against number of features patched and magnitude of edits for layers 3, 7, 11, 15, 19 and 23 of the residual stream of Pythia-410M. Pythia-410M has 24 layers, which we index 0, 1, ..., 23. Once again, we use $R = 4$ with varying $\alpha$ parameter.

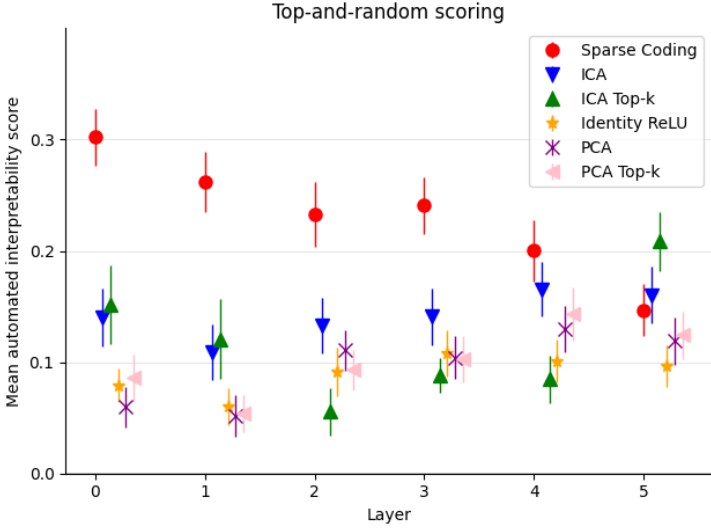

Figure 17: Autointerpretation scores across layers for the residual stream, including top-K baselines for ICA and PCA. As with Figure 3 these use $\alpha = 0.00086, R = 2$ and Pythia70M

