# OpenReview forum: "Sparse Autoencoders Find Highly Interpretable Features in Language Models"
_ICLR.cc/2024/Conference — ICLR 2024 poster_

### Official Review · Reviewer_apJf · 2023-10-29

**Soundness:** 2 fair
**Presentation:** 3 good
**Contribution:** 2 fair
**Rating:** 5
**Confidence:** 4

**Summary:**

The authors train a sparse autoencoder to find interpretable hidden units in an LLM. The sparse autoencoder linearly transforms the learned representation in a layer of the LLM and then applies a rectifying nonlinearity. The resulting code is regularized by L1 penalty so as to prefer sparse representations and it is trained to do well at reconstructing the LLM representation through a linear map. Using previously published methods to quantify the level of interpretability of a feature with a score,  the authors show that some of the resulting sparse features (those with the highest interpretability score) are more interpretable than those obtained with ICA, PCA and random projections. Other experiments study the effect of perturbing one of these sparse features on the output as well as whether some features are associated with a single word (monosemantic features).

**Strengths:**

Interpretability is an important subject for improving the safety of frontier AI systems, so this work belongs in a socially important research niche. The main contribution of showing greater interpretability with sparse features derived from the learned LLM layers is useful, although similar sparse representations have already been shown to be useful from an interpretability point of view for neural networks.

**Weaknesses:**

The paper claims in the abstract and conclusion that the results show *greater* monosemanticity than other methods but I do not see such comparisons in the paper (it should be in section 5.1). Either I missed something or one of the main claims is not supported.

The authors used linear encoders (followed by a ReLU) while it is well known that this is suboptimal (the optimal encoder needs to be much more nonlinear). A useful contribution would have been to evaluate and compare different types of encoding mechanisms for obtain a sparse representation. A simple variant is to introduce hidden layers (and depth) in the encoder. There are also optimization-based encoders that work very well and are well-known for sparse coding.

Most of the pieces in this paper (the sparse autoencoder and the interpretability score) are pre-existing, and so is the idea and demonstration that sparse features yield greater interpretability. This makes the overall level of originality and significance of the paper pretty low.

In the first paragraph of sec. 5.1, the authors talk about 'real-word features'. Please clarify what that means. If it is a "human-understable concept", one crucial problem is that such a concept (except for single-token cases studied in the paper) can be represented in many ways as a sequence of words, making any practical extension of the proposed idea (in sec 5.1) not obvious at all (or maybe not even feasible).

In section 5.3, it would be good to add evaluations using not just the top hidden layer (which has a very special semantics because it lives in the space of token embeddings) but also lower hidden layers

In section 6 about limitations, I would venture that two reasons for the poor reconstruction could be that (a) the proposed linear encoder does not have enough expressive power or (b) there exists no sparse linear basis to explain the LLM layers or (c) there is no compact natural language description for the sparse features.

Finally, the whole study is only looking at the *most interpretable* features in the lot, but in order to obtain any kind of safety guarantee, one will need to make sense of *all* the information in the LLM layers. We seem very far from that, which makes me doubt that the whole approach will ever be providing sufficient safety reassurances one day.

**Questions:**

Please try to address the questions I raised in the previous section.

---

> ### Author Response · Authors · 2023-11-16
> **Reply to Reviewer apJf**
>
> Our claim that we have greater monosemanticity rests on our comparisons in section 3 to the neuron basis, random directions, and directions found by PCA and ICA.
>
> On the originality of the approach, while we agree that none of the individual elements is novel on its own, the pipeline of using a sparse autoencoder to decompose activations in a large model (section 2), which are then passed to an automatic interpretation protocol (section 3), and then analysed in terms of the circuits that build up later features (section 5) represents a meaningful step in our ability to peer into the inner workings of language models.
>
> While we didn’t show these results, preliminary experiments using multilayer encoders did not show improvements in the reconstruction/sparsity curves and manual examination of the features found showed that they tended to be less interpretable, so we abandoned this line of work at an early stage.
>
> While there are optimization-based encoders, the iterative way that the coefficients are calculated means that they take significantly longer to train, we wanted an approach that is scalable to lots of data (which we found to be key to training these models) and larger models.
>
> We appreciate the point about ‘real-world features’ lacking clarity, and we’ve edited the text to amend this.
>
> We think your characterization that we’re “only looking at the most interpretable features” applies only to Section 5 of the paper, and not to “the whole study”. In particular, in Section 3 we compute the interpretability scores on randomly chosen features, and find our average score is much higher than that of our comparison methods. Additionally, in Section 4 our selection method for features is their impact on the IOI task, not any a-priori interpretability. You’re correct to say that a true safety guarantee would only be obtained by a complete enumeration of features, though we do reconstruct the vast majority of the variance. The most fundamental limitation appears to us to be the in-built assumption that features can be represented by directions and thus are linear. We agree that how much this assumption can be relied upon is uncertain and a very important direction for future work.
>
> > In section 5.3, it would be good to add evaluations using not just the top hidden layer (which has a very special semantics because it lives in the space of token embeddings) but also lower hidden layers
>
> The section 5.3 figure (previously Figure 5, now numbered Figure 6) does show results for the lower hidden layers. Namely, the diagram is generated recursively, where we start by with a layer 5 feature and identify its most causally-relevant layer 4 features, and then for each of those layer 4 features independently find the previous causally-relevant layer 3 features, etc.
>
> We thank you for your constructive feedback, and hope that our answers and updates have addressed your concerns. If you have other feedback or concerns, please let us know. If you think these changes have improved the paper, we hope you will update your scores to reflect that.

---

> > ### Comment · Reviewer_apJf · 2023-11-16
> > **partial interpretation problem**
> >
> > I have the impression that my comment on "only looking at the most interpretable features” was misunderstood. Let me try to clarify. Even if one looks at a random *subset* of features, most features are not interpreted. And doing better than the competition in terms of interpretability is great but still does not address the problem I had in mind, i.e., that a dangerous capability could still easily go unnoticed if we are not lucky in capturing it with one of our sparse features, either because it does not correspond to any of the sufficiently interpretable features, or because it would not be humanly feasible to "look" at a number of features comparable (or greater because of the sparsity) to the number of dimensions of the representation.
> >
> > To put things in perspective, a regulator is likely to have to take a decision about accepting the deployment of a future LLM which could greatly facilitate misuse in the form of national security threats. Can the regulator really rely on this kind of approach, is there a roadmap where this sort of approach would be sufficiently strong to reassure the regulator that catastrophic risks will be avoided?
> >
> > Of course, I did not expect this paper to solve the interpretability paper, which I understand is hard.
> > On the other hand, reflecting on the limitations and potential future workarounds or completely different approaches (if this approach may end up being a dead end) would greatly enhance the paper.

---

> > > ### Author Response · Authors · 2023-11-17
> > > **Reply to partial interpretation problem**
> > >
> > > Thank you for the clarification! You raise two important possibilities - that this technique simply isn’t good enough to capture all of the relevant features, and that even if it were, the number of features would be impractical.
> > >
> > > The first we agree is super important - we haven’t demonstrated that we can understand a full layer by this technique - though we think that we show sufficient improvement that we should be hopeful for progress in this area, whether by improved training techniques or alternative methods of feature identification.
> > >
> > > We see the second as less of an issue. We think that looking over such a large number of features might not be *humanly* feasible, but is going to be easily feasible for future generations of LLMs to do at scale. We didn’t have the level of funding to apply automatic interpretation to all of our found features but that’s less likely to be an issue for the large labs who are training these models or future government regulators. This scalability concern is a big reason why we used automatic interpretation, despite it being still at quite an early stage of development. We also think that the set of features to be interpreted can be narrowed significantly - for example if we want to use this technique for removing capacity to build bioweaponry, examining those features which are often active when working on these sorts of tasks should be significantly easier.
> > >
> > > Whether a regulator could have sufficient confidence in these techniques is difficult to judge - certainly such confidence would rely on advances to the technique combined with improvements in evaluations. We think that for goals like unlearning of dangerous knowledge, where you can perhaps directly remove its ability to process certain patterns, we could indeed have sufficient confidence that these capabilities are not present.
> > >
> > > We think that a bigger difficulty is to map from features to higher level behaviours - lots of dangerous behaviour won’t neatly map to single or small numbers of features and there will need to be progress in understanding the underlying generators of agentic behaviour. Nonetheless we hope that having a better lens into the internal workings of this model can speed up progress on this and other fronts.

---

### Official Review · Reviewer_k1em · 2023-10-30

**Soundness:** 3 good
**Presentation:** 2 fair
**Contribution:** 4 excellent
**Rating:** 6
**Confidence:** 4

**Summary:**

This paper presents a way to make the individual features of Large Language Models more interpretable by learning simple autoencoders with activation sparsity. They demonstrate that the learned features are more interpretable than original or simple methods like PCA via several ways, such as automated interpretability, example neurons and helpfulness in detecting circuit components.

**Strengths:**

Tackles a very important issues with mostly sound methodology and original results. The improvement in interpretability is quite noticeable, and evaluation is diverse.
Figure 4 is a good clear explanation of 1 feature, and the comparison to the relevant feature in standard residual stream is very useful.
Figure 5 is also interesting and a good example of the potential usecases of this method.

**Weaknesses:**

Missing a lot of important basic information.
 - What is R? It is an important part of many text captions such as Figure 2 and Figure 3 but never explained in main text. Only explained in Appendix C.
- Almost all the results do not mention the details of how they were achieved. In particular, table 1, and figures 2, 4, 5 don't mention the model used at all, and most figures do not mention R and alpha used to train the in question.

In general the main text needs more discussion of the basic methods and results like reconstructions loss, some of the content from Appendix B and C should be moved to main text.

The high reconstruction loss seems like a problem for the usefulness/faithfulness of this method, and I would like more discussion on it, such as the i.e. large increase in perplexity discussed in section 6.2. What R and Alpha were used for this result? How does this change as a function of different parameters like R and alpha or different layers?

**Questions:**

- Figure 2: Is identity ReLU the default basis?
- Section 3.2: Why is ReLU applied to the default and random basis? It is said its applied to make activations nonnegative, but why is this necessary? The actual model also uses negative activations right?
- What is the intended way to use these learned features? Adding the autoencoder changes the network behavior, and increases computational cost. Are you supposed to use the expanded network in practice to make explainable decisions, or is this better thought as an approximation to explain decisions made by the unmodified network.

---

> ### Author Response · Authors · 2023-11-16
> **Reply to Reviewer k1em**
>
> We appreciate your call for more clarity on how results have been achieved! All tables and figures now indicate which model and hyperparameter values were used.
>
> We think the point that some of the material in Appendices B and C should be in the main text is an interesting one. We’ve brought some of Appendix B forward into a new Figure 2 which hopefully gives the reader better insight into the nature of the models that we trained. While the material in Appendix C is interesting, we feel that the fact that the learned bases are under-complete in many cases represents a significant drawback to them that wouldn’t be appropriate to show in the main text, and which is best shown in the appendix as a very enticing but ultimately incomplete direction for future work to build on. (We also value the range of experiments and don’t have pages to spare!)
>
> Answers to questions:
>
> > What is R?
>
> R is defined in the second paragraph of section 2: “The autoencoder is a neural network with a single hidden layer of size d_hid = R d_in, where d_in is the dimension of the language model internal activation vectors1, and R is a hyperparameter that controls the ratio of the feature dictionary size to the model dimension.”
>
> > Figure 2: Is identity ReLU the default basis?
>
> Yes
>
> > Section 3.2: Why is ReLU applied to the default and random basis? It is said its applied to make activations nonnegative, but why is this necessary? The actual model also uses negative activations right?
>
> We restricted all activations to be positive here because we were concerned that negative activations were a source of additional variance that would bias the results in favour of our method. The interpretation score is based on a correlation metric and thus has variance in the denominator. If an explanation could explain the same amount of variance in the positive domain for one of our features and one default basis features, the resulting correlation score would be higher for our feature, thus biasing our results. In our early experiments we found that autointerpretability scores were similar with or without ReLU. It is likely that some features are bidirectional and so would be best explained by allowing a feature to have both positive and negative activations, but this would be penalized equally for our features and default/random/PCA features.
>
> > What is the intended way to use these learned features? Adding the autoencoder changes the network behavior, and increases computational cost. Are you supposed to use the expanded network in practice to make explainable decisions, or is this better thought as an approximation to explain decisions made by the unmodified network.
>
> The latter is definitely more what we have in mind. A core principle of this research is to find interpretability techniques that don’t come at a performance cost - we want to be able to run the model as normal and use this ‘featurized’ version of a layer to understand what the model is doing as best we can, hopefully eventually being able to detect behaviour that’s deceptive or otherwise not in line with the developer’s intentions. We also think that the utility of the current technique is as a platform for further work in understanding how models compute their outputs - we think we should eventually be able to learn better features, and also to build them up into more interesting units like circuits for particular behaviours.
>
> Another hypothesis, though as yet unproven, is that they may be useful for model editing. For example, model editing often involves learning a vector to add to a residual stream, but finding the right direction across dozens of 10K+ dimensional spaces requires quite a bit of data and these features could provide a useful prior for productive directions to edit.
>
> We appreciate the helpful comments and questions and we hope this makes the presentation better. If it seems markedly improved we’d really appreciate it if the score were updated to reflect that, else if there are further changes you’d recommend please let us know.

---

> > ### Comment · Reviewer_k1em · 2023-11-23
> > **Thanks for the response**
> >
> > This has addressed several of my concerns regarding clarity, and I believe the paper is stronger than before. However, it would be useful to have more transparency around the changes to the manuscript during discussion period such as using a different color for changes in the manuscript.
> >
> > In addition, some of my concerns were still unaddressed such as expanded discussion on perplexity in section 6.2. In fact, it seems you have done the opposite and removed it from the paper all together, why is this the case?
> >
> > As a result I will retain my original rating.

---

### Official Review · Reviewer_oRDV · 2023-10-31

**Soundness:** 1 poor
**Presentation:** 1 poor
**Contribution:** 1 poor
**Rating:** 1
**Confidence:** 5

**Summary:**

This paper is addressing the challenge of polysemanticity in DNNs. In DNNs context neurons appear to activate in multiple, semantically distinct contexts. This paper is trying to address the challenge of disambiguating neurons. They suggest sparse auto-encoders for this challenge.

**Strengths:**

+ Very interesting research question

**Weaknesses:**

- Overly complex approach, where the auto-encoder will likely add noise to the results
- No real baseline comparison
- Experimentation is weak
- Results are difficult to interpreter and would need to be better presented or colored in the context

**Questions:**

N/A

---

> ### Author Response · Authors · 2023-11-16
> **Reply to Reviewer oRDV**
>
> We’re sorry that you didn’t find the paper of value though we hope that the other reviews which recommend acceptance suggest that there’s content which is informative for others.
>
> We agree that interpretability is a difficult area to find real benchmarks in, and the benchmarks are not as robust as would be expected in most areas of ML, though through the use of automatic interpretability we have tried hard to give performance measurements which have clear definitions and can be replicated. We note that Bills et al represents a core push by OpenAI to understand their models and so being able to find primitives which are advances according to their own protocols represents a significant step among those working in this area.
>
> We disagree that the approach is overly complex - to us it is both simple and directly motivated by the theoretical considerations in Toy Modes of Superposition. The computational simplicity of the linear autoencoder over iterative methods for sparse coding (which you may be implicitly referring to as the less complex approach?) is likely to be very important in scaling this technique up to cutting edge models.

---

> > ### Comment · Reviewer_oRDV · 2023-11-20
> > **Thanks for your rebuttal**
> >
> > Thank you for your rebuttal. It would be nice to have some factual figures to back up: *The computational simplicity of the linear autoencoder over iterative methods for sparse coding (which you may be implicitly referring to as the less complex approach?) is likely to be very important in scaling this technique up to cutting edge models.* Would you have such evidence?

---

> > > ### Author Response · Authors · 2023-11-23
> > > **Evidence of Computational Simplicity**
> > >
> > > Yes, a collaborator recently completed a comparison experiment between a sparse autoencoder and the FISTA method (as used in Yun et al https://arxiv.org/abs/2103.15949). In this comparison experiment, they trained 8 dictionaries on 512-dimensional data, with a feature ratio R=1, on a 1050 Ti. The results: autoencoders completed training in ~15 minutes, compared to ~10 hours for the FISTA dictionaries. In this experiment at least, the autoencoder was 40x faster than FISTA.

---

### Official Review · Reviewer_FVSC · 2023-11-01

**Soundness:** 3 good
**Presentation:** 3 good
**Contribution:** 3 good
**Rating:** 6
**Confidence:** 4

**Summary:**

This paper proposes using sparse autoencoders to learn interpretable and monosemantic features from the internal activations of language models. The key idea is to reconstruct model activations as sparse linear combinations of latent directions, which aims to disentangle the model's features from superposition. The authors train autoencoders with an L1 sparsity penalty on various model layers and analyze the resulting feature dictionaries.
* The learned dictionary features are shown to be more interpretable than baseline methods like PCA/ICA according to automated scores.
* The features localize target model behaviors more precisely for a counterfactual evaluation task.
* Case studies demonstrate some highly monosemantic features that influence model outputs in predictable ways.
* The approach is scalable and unsupervised, requiring only unlabeled activations.

**Strengths:**

Applies sparse coding in a novel way for model interpretability. The approach is intuitive and theoretically motivated.
Provides solid evidence that the learned features are more interpretable and monosemantic.
Demonstrates the technique can pinpoint features for analyzing model behaviors.
Case studies show highly intuitive individual features.
The method is scalable and unsupervised.

**Weaknesses:**

The reconstruction loss is not fully minimized, indicating the dictionaries do not capture all information.
The method works less well for later model layers and MLPs compared to early residual layers.
More analysis needed on:
1) generalizability of behaviors across tasks,
2) sparsity of dependencies between features.

Limited comparison to other interpretability and disentanglement techniques.  For example, beta-VAE, infoGAN, FactorVAE had similar disenanglement goals.  And Michaud 2023 "The Quantization Model of Neural Scaling" suggested a spectral clustering approach for identifying "monogenic" signals in large models. The paper would be strengthened if it provided a more complete comparison to previous disentanglement approaches.

The authors could mention the previous works in the field of sparse autoencoding and dictionary learning for word embeddings. Subramanian et al. [1] similarly found linear factors for word embeddings, in this case using a sparse autoencoder. Zhang et al. [2] solved a similar problem using methods from dictionary learning. Their method discovers elementary structures beyond existing word vectors.

[1] Spine: Sparse interpretable neural embeddings Subramanian, A., Pruthi, D., Jhamtani, H., Berg-Kirkpatrick, T. and Hovy, E., 2018. Proceedings of the AAAI Conference on Artificial Intelligence, Vol 32(1).

[2] Word embedding visualization via dictionary learning Zhang, J., Chen, Y., Cheung, B. and Olshausen, B.A., 2019. arXiv preprint arXiv:1910.03833.

**Questions:**

How well do the features transfer across different models and architectures?
Could you incorporate model weights or adjacent layer features to improve dictionary learning?
What changes allow learning overcomplete MLP bases?
How well do the features generalize to unseen tasks? More thorough evaluation would be useful.
The authors claim the method is scalable, but have not demonstrated it on very large models. Experiments on models with billions of parameters could better support the scalability claims.

---

> ### Author Response · Authors · 2023-11-16
> **Reply to Reviewer FVSC**
>
> We agree that the autoencoder’s imperfect reconstruction is a drawback, but getting much higher auto-interpretation scores still marks this out as a significant step towards being able to understand a layer in a large language model.
>
> We think that betaVAEs are interesting points of comparison though their motivation and structure are sufficiently different that they can’t be directly compared to our approach.
>
> The work from Subramanian et al and others working on word embeddings is indeed an important precursor to our work that we weren’t aware of, and we really appreciate these comments and have updated the related works section accordingly.
>
> We think all of your questions are interesting points and look a lot like our plans for future work, especially incorporating model weights and trying harder to learn overcomplete bases in the residual stream (we suspect much longer training times and reinitialization are likely both to be key). As for scalability, we note that our subject models, Pythia70M and less often Pythia410M are larger than other examples, with the largest being Yun et al using a BERT model with ~100M parameters. The fact that we use autoencoders rather than iterative methods such as FISTA makes scaling significantly easier as only a single pass is needed to compute the coefficients. Nonetheless, the compute required is quadratic in the width of the layer (and possibly even larger if larger ratios and longer training runs are required), so scaling to multi-billion parameters while maintaining quality models requires engineering beyond the scope of this paper.
>
> Replies to questions:
>
> > How well do the features transfer across different models and architectures?
>
> We did not test this, but there is no a-priori reason to expect these features to transfer, since they are just linear directions and a model’s residual stream is (in theory) linear, so therefore two models will have “arbitrarily transformed” residual streams relative to each other.
>
> > Could you incorporate model weights or adjacent layer features to improve dictionary learning?
>
> While we have not attempted this, a recent work (https://www.lesswrong.com/posts/YJpMgi7HJuHwXTkjk/taking-features-out-of-superposition-with-sparse) has explored using model information to initialize the dictionaries to improve training speeds.
>
> > What changes allow learning overcomplete MLP bases?
>
> Could you clarify what you’re asking here?
>
> > How well do the features generalize to unseen tasks?
>
> Both the Pythia models and our autoencoders are trained on The Pile, a large text corpus, so all are equally seen (or unseen) up to their relatively frequencies in the corpus.
>
> We thank you for your constructive feedback, and hope that our answers and updates have addressed your concerns. If you have other feedback or concerns, please let us know. If you think these changes have improved the paper, we hope you will update your scores to reflect that.

---

> > ### Comment · Reviewer_FVSC · 2023-11-20
> >
> > > > What changes allow learning overcomplete MLP bases?
> >
> > > Could you clarify what you’re asking here?
> >
> > I'm asking about your findings when applying your methods to MLPs, e.g., Appendix C.3.  You have mentioned that your method learns a lot of dead features and never is able to learn an overcomplete basis of actual live features.
> >
> > Particularly:
> > * Why does this happen - i.e., why doesn't your method not work well from MLP fan-out layers?  You've noted the problems and we're asking "why."  These seem like the most promising layers to understand.
> > * Do you have any insights on why Eq5 and 6 worked on earlier layers, and what is your hunch on how to resolve the lower performance on later layers?

---

> > > ### Author Response · Authors · 2023-11-22
> > > **Difficulties with MLP layer**
> > >
> > > These are good questions, and while we’ll describe our best current theories, we definitely don’t know for sure. This uncertainty is a big reason why these results are included as an appendix and we’d be excited to see future work that shines more light on this.
> > >
> > > Our the three best answers to your questions are:
> > > - There just isn’t a good overcomplete basis in these layers and we need a different framework to make sense of these later layers.
> > > - There is a good overcomplete basis, but the geometry of MLP layers makes learning it hard, which we need to reflect in our training dynamics somehow.
> > > - There is a good overcomplete basis and we just need to train for much much longer.
> > >
> > > We’ll go through these in more detail.
> > >
> > > Firstly, as you note, we did find overcomplete bases in the early layers - it’s the later layers which are the problem. This mirrors the fact that improvement in reconstruction loss in the residual stream from doubling the width is much higher in the early layers, so it may be the case that the typical geometry of those layers allows them to better be captured by a smaller number of ‘features’ and/or makes the long tail of features harder to find. This doesn’t mean that there can’t be useful information from learning a high quality overcomplete basis, just that it seems there’s something fundamental about later layers that makes this more difficult. Also, the output of the non-linearity in a GPT-style transformer is 4x wider than the residual stream, but is only a relatively simple transformation of the residual stream, so we might expect to need a smaller ratio R in the MLP. Still, since the reconstruction loss is significant here, we should not be complacent and must continue to look for ways to understand these layers. Perhaps more complicated encoders are needed and we just failed to get them to work, or perhaps we need something more flexible than discrete linear features.
> > >
> > > The second possibility is that the MLP’s geometry interferes with our training process. In particular, MLP activations are mostly in the positive orthant of activation space, leading to more dead neurons in the training process. We tried a couple alternative approaches to overcome this but they were unsuccessful. In particular, we tested our technique on the MLP layer with: 1) centered data, and 2) adding the minimum of the GELU function to all activations and then enforcing only positive entries in the weight matrix. It’s still possible that this is a major cause but our attempts to address it weren’t successful.
> > >
> > > Our final hypothesis is that we simply need to train for orders of magnitude longer, potentially in conjunction with reinitializing dead neurons. In this answer we take inspiration from a recent Anthropic paper (https://transformer-circuits.pub/2023/monosemantic-features/) which used a technique similar to ours on the MLP layer of a one-layer transformer. They used hundreds of times more examples in training, along with reinitialization, and found that they were able to learn a very highly overcomplete basis. Since their model had 512 MLP dimensions rather than our 2048, this suggests we should try increasing our training time by perhaps 1000x. However, the fact that their language model has only a single layer leaves some uncertainty about whether these techniques assist here: is that single layer most analogous to our layer 0 (where our existing technique work) or our layer 5 (where problems arise)? Finally, in our experiments we chose not to extend our training runs because the marginal improvement seemed to be minimal. In particular, we found that at the upper end of our training time, the number of commonly active features was still increasing, but at a very slow rate such that it would not meaningfully change with another doubling of training time, so we didn’t push further.

---

> > > > ### Comment · Reviewer_FVSC · 2023-11-23
> > > >
> > > > Thanks for the detailed discussion of the issues and possibilities for future work.

---

### Official Review · Reviewer_V2wZ · 2023-11-09

**Soundness:** 3 good
**Presentation:** 3 good
**Contribution:** 3 good
**Rating:** 6
**Confidence:** 3

**Summary:**

Superposition refers to the hypothesis that neural model representations are in fact linear compositions of sparse features. This paper attempts to identify these sparse features using dictionary learning. More specifically, the authors train an autoencoder with sparsity penalties on language model representations of interest (as both input and output), and use the learned sparse encoding as a proxy to understand the original language model. The resulting sparse encodings are passed through an auto-interpretation model to identify meaningful features. Under the auto-interpretation evaluation, the learned sparse encoding by the proposed method beat several baselines.

**Strengths:**

The paper is well written and easy to follow. The idea of using sparse autoencoder to tackle superposition makes intuitive sense, and it turns out that it works surprisingly well. The activation patching experiments validates the effectiveness of the proposed method beyond the auto-interpretation metrics, which emphasizes the method's practical value.

**Weaknesses:**

Using dictionary learning for superposition isn't particularly novel.
Auto-interpretation score hasn't been shown to correlate with actual interpretability use cases yet.
Activation patching experiment was only conducted on one relatively synthetic dataset.

**Questions:**

Any comments on how the proposed method scales with complexity of the underlying task / number of possible features? Is the approach bottlenecked by auto-interpretation method?

---

> ### Author Response · Authors · 2023-11-16
> **Reply to Reviewer V2wZ**
>
> We’re glad you found it well written and the results surprisingly positive!
>
> We agree that it would be very valuable to see more work tying automatic interpretation to more concrete results and we acknowledge some of the limitations in the auto-interpretation section. We still think that its underlying method (measuring interpretability by the correlation between the actual activation of a neuron or feature and the activation as estimated from a potential description of the meaning of that feature) is well-founded, and we hope that by giving people better primitives than neurons to work with, we’ll be able to see more foundational work in this area soon.
>
> While the activation patching experiment was only conducted on a single, synthetic dataset, the IOI task that we chose is the most studied example of an LLM circuit that we’re aware of. It was therefore the first thing that we tried, so we think this demonstrates the practicality of the method, as you mention.
>
> > Any comments on how the proposed method scales with complexity of the underlying task / number of possible features?
>
> One of the big benefits of the approach is that it’s task agnostic - ideally a dictionary should contain all the relevant features - we certainly can’t claim that we reached this point we found all of them, if that’s even a well defined notion, but we don’t see any fundamental blocker to capturing features relevant to any task the model is capable of in a single training run.
> The method scales somewhere above linearly with the number of features - if you double the width of the autoencoder, you’ve twice as many parameters and there’s some additional slowness to converge though we didn’t try to quantify exactly how much extra time is needed.
>
> > Is the approach bottlenecked by auto-interpretation method?
>
> At the scale of this paper (Pythia70M, ~100M datapoints of training) it’s easier to train lots of autoencoders than to evaluate lots of them via automatic interpretation - a single feature evaluated using the procedure from Bills et al cost around £0.20 so getting a good sample size of around 200 features interpreted cost about £40 in August 2023 OpenAI credits, significantly more than the training cost per encoder (and meant that we couldn’t run automatic interpretation on most of the dictionaries we trained). Training the autoencoder is more than quadratic in the width of the layer being examined though, but interpretation is unaffected by layer width, so at larger scales the cost of the autoencoder will dominate.
>
> We thank you for your constructive feedback, and hope that our answers and updates have addressed your concerns. If you have other feedback or concerns, please let us know. If you think these changes have improved the paper, we hope you will update your scores to reflect that.

---

### Author Response · Authors · 2023-11-16
**Reply to all Reviewers**

We’d like to thank the reviewers for their insightful feedback. We have uploaded a revised version of the paper building on some of their suggestions. Some themes arose across multiple reviews, which we’d like to address here.

 **What is novel here?**
Our reviewers correctly noted that our work follows the rich tradition of learning a dictionary to sparsely represent semantic information. We especially appreciate Reviewer FVSC pointing us to papers applying similar techniques to word embeddings, which we have now incorporated into an expanded Related Works section. Given this tradition, we want to emphasize what we see as novel in this paper, including novel combinations of previously-pioneered techniques.

Namely, we present:
- A sparse autoencoder architecture, similar to Subramanian et al and building on the preliminary work of Sharkey et al.
- Applying this architecture to decompose residual stream activations, similar to Yun et al.
- Using the techniques of Bills et al to systematically measure the interpretability of all these features.
- That our technique improves on this metric over the baselines of the neuron basis and PCA.
- We synthesize several more analyses to show the improved interpretability of our features:
  - We provide empirical evidence that a particular behaviour (performance on the IOI task) is localized to a smaller number of features compared to baselines.
  - We find individual features with clear function based on the tokens that activate the features and the effect on output token predictions.
  - We trace a circuit of causal dependencies between these features.
We think the combination of these techniques provides a novel approach to attribution which is more than the sum of its parts.

**Our approach is task-agnostic**

Reviewers ​​V2wZ and FVSC ask questions about the task either underlying or unseen by the autoencoder. We wish to emphasize that our autoencoders were trained in a task-agnostic way, via randomly-sampled text from The Pile, a large webtext corpus. Thus there is no particular emphasis placed on any one task over another in the training of our autoencoders (and the same is true for the Pythia language models we study, which are trained to do next-token prediction on The Pile). When we narrow our focus to the IOI task in Section 4, we use the same “general purpose” feature dictionaries as elsewhere in the paper.

**Imperfect reconstructions**

Reviewers FVSC, k1em, and apJf all note the autoencoder’s relatively high reconstruction loss, indicating that the sparse representation we found may fail to capture some significant information from the model. However, given the difficulty of finding explanations for neurons demonstrated in Bills et al, the fact that we are able to generate significantly better explanations for our features while capturing over 90% of the variance is an important step in the right direction.

---

### Meta-Review · Area_Chair_Hf6L · 2023-12-06

**Metareview:**

The paper proposes to use sparse autoencoders (with a single hidden layer) to uncover the latent features in language models. The goal is to discover the true unit of computation, which can then be used to build other interpretability techniques, such as perform automated circuit discovery. The authors show that the proposed method discovers more interpretable features than the baselines such as PCA, and that it can be used to do automatic circuit discovery in a case study.

The question of identifying the true latent features and addressing superposition / polysemanticity is a core challenge in mechanistic interpretability. For example Anthropic recently released a related [study](https://transformer-circuits.pub/2023/monosemantic-features) which also uses single-layer autoencoders to discover features in LLMs, although they focus on much smaller models. This concurrent work demonstrates the relevance of the problem.

## Weaknesses

Several reviewers pointed out that there are still major limitations to the autoencoder approach. In particular, the reconstruction loss is imperfect. It is unclear if the true features can be captured with a single-layer autoencoder. It is unclear if enumerating all features and considering all of them is a viable strategy for AI safety.

Scientifically, interpretability lacks clear metrics generally, so it is hard to evaluate proposed methods and compare to baselines. The study is partially qualitative.

Several reviewers noted that this work builds on prior work, and there are several related papers.

## Strengths

I believe, this paper is making a significant contribution to the field of mechanistic interpretability. It demonstrated that autoencoders provide a better feature representation than naive baselines that people have previously used. In particular, the improvements in automated interpretability scores are significant and promising.

Importantly, the method can be scaled to much larger models. It is a simple linear autoencoder that can be scaled up significantly. The simplicity of the method is an advantage from this perspective.

**Justification For Why Not Higher Score:**

The reviews are generally mixed, and none of the reviewers champions for the paper. There are several concerns from reviewers which were not fully addressed.

**Justification For Why Not Lower Score:**

I believe, this paper makes an important contribution to the field of mechanistic interpretability. Many of the reviewers were positive about the paper, and the remaining limitations are generally hard to fully address within a single paper. I encourage the authors to try to accommodate the remaining feedback from the reviewers in the next version of the paper.

---

### Decision · Program_Chairs · 2024-01-16

Accept (poster)